# Lack of asmt1 or asmt2 Yields Different Phenotypes and Malformations in Larvae to Adult Zebrafish

**DOI:** 10.3390/ijms26083912

**Published:** 2025-04-21

**Authors:** Paula Aranda-Martínez, José Fernández-Martínez, María Elena Díaz-Casado, Yolanda Ramírez-Casas, María Martín-Estebané, Alba López-Rodríguez, Germaine Escames, Darío Acuña-Castroviejo

**Affiliations:** 1Centro de Investigación Biomédica, Facultad de Medicina, Departamento de Fisiología, Instituto de Biotecnología, Parque Tecnológico de Ciencias de la Salud, Universidad de Granada, 18016 Granada, Spain; ampaula@correo.ugr.es (P.A.-M.); josefermar@ugr.es (J.F.-M.); yolandaramirez@ugr.es (Y.R.-C.); mmestebane@ugr.es (M.M.-E.); albalopezrodriguez4@gmail.com (A.L.-R.); gescames@ugr.es (G.E.); 2Instituto de Investigación Biosanitaria (Ibs. Granada), Hospital Clínico Universitario San Cecilio, 18016 Granada, Spain; 3Centro de Investigación Biomédica en Red de Fragilidad y Envejecimiento Saludable (CIBERFES), Instituto de Salud Carlos III (ISCIII), 28029 Madrid, Spain; 4UGC de Laboratorios Clínicos, Hospital Universitario Clínico San Cecilio, 18016 Granada, Spain

**Keywords:** zebrafish, asmt1, asmt2, melatonin, characterization

## Abstract

Melatonin is an indolamine derived from tryptophan, which is highly conserved throughout evolution, including in zebrafish, where it controls important cellular processes, such as circadian rhythms, oxidative stress, inflammation, and mitochondrial homeostasis. These functions of melatonin and its synthesis route are quite similar to those in humans. One of the most important enzymes in melatonin synthesis is acetylserotonin O-methyltransferase (ASMT), the rate-limiting enzyme, which catalyzes its final step. Due to genome duplication, zebrafish has two genes for this enzyme, asmt1 and asmt2. These genes show differential expression; asmt1 is primarily expressed in the retina and the pineal gland, and asmt2 is expressed in peripheral tissues, indicating different functions. Therefore, the aim of this work was to develop a mutant model for each asmt gene and to analyze their phenotypic effects in zebrafish. The results showed that the loss of 80% of the asmt2 gene affected melatonin concentration and consequently disrupted the sleep/wake rhythm in larvae, decreasing by 50% the distance traveled. In contrast, the loss of asmt1 had a greater influence on the physical condition of adults, as locomotor activity decreased by 50%, and 75% showed malformations. These data reveal distinct functional roles of melatonin depending on their site of production that may affect the development of zebrafish.

## 1. Introduction

Melatonin, or N-acetyl-5-methoxytryptamine (aMT), is an indolamine derived from tryptophan, which is highly conserved throughout evolution [1]. Initially, it was thought to be produced only in the pineal gland, but it is now known to be produced in most organs and tissues of the body [2]. Therefore, two types of melatonin are distinguished: pineal and extrapineal melatonin. The former is released into circulation in a circadian manner, reaching all organs and tissues to exert a chronobiotic function. Extrapineal melatonin, in turn, is produced at higher concentrations than the pineal one, but it does not have a production rhythm and remains within the cell, not entering into circulation. It acts mainly to maintain mitochondrial homeostasis [3]. Moreover, extrapineal melatonin acts as a potent antioxidant by directly eliminating free radicals and enhancing antioxidant defense and as an anti-inflammatory molecule by modulating the innate immune response [2]. Thus, a reduction in melatonin leads to an increase in oxidative stress, inflammation, mitochondrial dysfunction, and apoptosis, along with a disruption of circadian rhythms. These physiological processes are characteristic of many pathologies, such as obesity and diabetes [4], neurodegenerative diseases [5], sarcopenia [6], and cardiovascular diseases [7], among others.

In zebrafish, both the functions and the synthesis pathway of melatonin are highly similar to those in humans. The most important enzymes in melatonin synthesis are aralkylamine N-acetyltransferase (AANAT) and acetylserotonin O-methyltransferase (ASMT). ASMT catalyzes the final step of melatonin synthesis and is also the rate-limiting enzyme1. Zebrafish has two genes for this enzyme, asmt1 and asmt2, due to the genome duplication specific to teleost fish [8]. It has been observed that both genes are highly conserved in teleost fish, maintaining a similar exon–intron pattern and showing high sequence similarity with other vertebrates, including humans [9]. However, differential expression was found between the two genes, where asmt1 was primarily expressed in the retina and the pineal gland and asmt2 was expressed in peripheral tissues, such as the liver, intestine, skin, and gonads. These results, along with some variations in residues between the asmt1 and asmt2 sequences, seem to indicate different functions [9].

Given the differential expression of asmt1 and asmt2 in zebrafish and their structural divergence, as well as the lack of studies about asmt mutations, it is essential to elucidate whether these genes have distinct physiological roles. Zebrafish is an ideal model to investigate melatonin-related functions due to its genetic tractability, conserved melatonin pathway, and clearly defined circadian and developmental phenotypes [10]. Despite this, no study has yet compared the phenotypic consequences of asmt1 and asmt2 loss of function. Therefore, understanding the specific roles of each gene will shed light on tissue-specific melatonin functions and their implications for development and aging.

## 2. Results

### 2.1. Drop in Melatonin Level Is Higher in asmt2 than in asmt1 Mutants

After developing the mutant zebrafish models, we first decided to analyze whether the mutation in asmt1 and asmt2 affected the concentration of melatonin. We observed that melatonin levels similarly decreased by 50% in asmt1 heterozygous mutants (asmt1^+/−^), asmt1 homozygous mutants (asmt1^−/−^), and asmt2 heterozygous mutants (asmt2^+/−^) compared with the WT group (Figure 1). However, melatonin concentration decreased significantly by 80% in the asmt2 homozygous (asmt2^−/−^) mutants compared to the WT and by 30% compared with the other mutant lines (Figure 1).

### 2.2. Sleep/Wake Rhythm Is Disrupted by Mutations in asmt1 and asmt2

Because melatonin regulates the sleep/wake rhythm in zebrafish [11] and given the reduction of melatonin content in mutant zebrafish, we next studied the rhythm of motor activity in these animals during the day and the night. The WT group showed a typical daily rhythm, with activity starting when the lights were on and decreasing at night, along with high motor activity during the day (Figure 2A). Although asmt1^+/−^ and WT traveled similar distances (Figure 2F–H), the former displays a significant phase advance of its activity that started in the dark phase (Figure 2B). On the other hand, asmt1^−/−^, asmt2^+/−^, and asmt2^−/−^ exhibited nearly flat and arrhythmic activity (Figure 2C, Figure 2D, and Figure 2E, respectively). The distance traveled by asmt1^−/−^ and asmt2^+/−^ decreased during the day (Figure 2G) but not during the night (Figure 2H). The most affected group was asmt2^−/−^, which showed a significant reduction in the distance traveled during the day and the night (Figure 2G,H), which was also reflected in the total traveled distance compared with the other groups (Figure 2F).

### 2.3. Physical Activity Was More Compromised in the asmt1 Homozygous Zebrafish than in the asmt2 Group

Recently, it has been demonstrated that chronodisruption affects muscle function and, therefore, physical activity [12], and because the loss of melatonin causes chronodisruption [5], we decided to study how the alteration of the asmt genes affects physical activity in zebrafish. The asmt1^+/−^ group covered a total distance similar to that of the WT, with no significant changes. In contrast, we observed a significant reduction in asmt1^−/−^ compared to them and also in the asmt2 mutant lines, which were similar to the WT (Figure 3A). Similarly, asmt1^+/−^ and asmt2^+/−^ showed a maximum speed comparable to the WT group. In this case, the maximum speed decreased slightly in both homozygotes, which was only significant in asmt2^−/−^ compared to asmt1^+/−^ (Figure 3B). The mean speed was only significantly reduced in the asmt1^−/−^ group, while asmt1^+/−^, asmt2^+/−^, and asmt2^−/−^ remained similar to WT (Figure 3C). Finally, asmt1^−/−^ exhibited a significantly longer resting time than WT, asmt1^+/−^, asmt2^+/−^, and asmt2^−/−^ (Figure 3D).

### 2.4. Mutation in asmt1 but Not in asmt2 Produced Morphological Changes in Zebrafish Larvae

It is known that melatonin regulates cell proliferation during zebrafish development [13], so we decided to analyze some morphometric aspects in the larvae to observe the effect of the mutations in both genes on the larva phenotype. The total length of the larva was significantly shorter in both heterozygotes and homozygotes compared to the WT; however, the asmt1^+/−^ group experienced a more drastic reduction than the other mutants (Figure 4A). The asmt1^+/−^ group also showed a significant reduction in the width of the larva but not the other mutant lines (Figure 4B). Both the length of the eye and the length of the ear were significantly reduced in asmt1^+/−^ compared to the rest of the groups (Figure 4C,D). Finally, we also observed a significant reduction in the size of the swim bladder in the asmt1^+/−^ group (Figure 4E).

### 2.5. The Mutant Zebrafish Lines Showed Malformations During the Larval Stage That Increased in Adult Zebrafish

Because melatonin controls larval development and it has been observed that zebrafish develop malformations during the larval stage in other pathologies [14], we decided to study the influence of asmt1 and asmt2 on the development of larval malformations. WT zebrafish developed normally, without malformations. asmt1^+/−^ larvae yielded slight edema compared to the WT, whereas the absence of asmt1^−/−^ showed three types of malformation, including edema and tail and yolk abnormalities. As with WT, asmt2^+/−^ larvae showed a lack of malformations, and the absence of asmt2 only showed tail malformations (Figure 5A). The statistical data regarding these changes are shown in Appendix A.

To assess whether these malformations are maintained in adult fish, they were analyzed at the age of 2 years old. The WT group showed a small percentage of zebrafish with scoliosis, whereas asmt2^+/−^ remained absolutely normal. asmt1^+/−^ zebrafish showed scoliosis and caudal fin abnormalities, and the absence of asmt1 resulted in more severe scoliosis and caudal fin abnormalities. In turn, asmt2^−/−^ showed scoliosis, although to a smaller degree than in asmt1^−/−^, but a greater increase in caudal fin malformation (Figure 5B). Statistical data are reported in Appendix A.

## 3. Discussion

In vertebrates, including birds, fish, and mammals, ASMT plays a crucial role in melatonin synthesis and the regulation of circadian rhythms [1,15,16]. In addition to its central role in the biosynthesis of melatonin, asmt mutations have been associated with neuropsychiatric conditions, such as depression, anxiety, and bipolar disorder, as well as the maintenance of gut microbiota plasticity [17,18,19]. However, ASMT remains relatively understudied, and due to the conserved and essential role of ASMT in regulating physiological homeostasis, this supports the relevance of studying asmt1 and asmt2 mutations in zebrafish as a model for understanding melatonin-related functions.

In this study, the first asmt1 mutant zebrafish line has been developed, and a comparative study with the previously developed asmt2 mutant line has been conducted. We have performed a phenotypic characterization of both models in order to elucidate the role of asmt1 and asmt2 in the phenotypic characteristics of zebrafish.

First, we ensured that both mutants had a deficiency in melatonin production, which was reduced by more than half compared to the WT group in zebrafish larvae, but it was not null because in each case the other asmt gene is not mutated. However, the concentration of melatonin was more affected by the loss of asmt2 than asmt1, which may be due to the differential expression of both genes in zebrafish. The gene asmt2 has been shown to be more strongly expressed in peripheral tissues, which would result in a greater loss of melatonin compared to asmt1, which is only expressed in the retina and the pineal gland. The reduction in melatonin concentration affected the locomotor activity of the larvae, where we observed disruptions in the sleep/wake rhythm in all mutants, as melatonin controls circadian rhythms [11]. As expected, the asmt2−/− group was the most affected due to the drastic reduction in melatonin.

Additionally, melatonin plays a role in the larval development of zebrafish because it regulates cell proliferation in the larva [13], so we decided to analyze various phenotypic aspects of the larvae. Regarding the morphometric parameters of the larvae, almost no differences were found, except for the asmt1 heterozygote. The most visible morphometric changes in heterozygotes may arise from genetic phenomena, such as allelic imbalance, genetic redundancy, haploinsufficiency, and dominant-negative effects, whereas homozygotes may exhibit more severe functional deficits that are masked by compensatory mechanisms in the short term. On the other hand, malformations produced in zebrafish larvae due to the lack of asmt1 or asmt2 were evaluated, and a higher incidence of malformations was observed in the homozygotes for both genes.

Due to the fact that melatonin decreases with age [20], affecting circadian rhythms, oxidative stress, inflammation, and many other cellular processes closely related to aging [21], we also analyzed the loss of asmt1 and asmt2 in adult zebrafish. The deficiency in asmt2 led to an increase in malformations in the caudal fin and scoliosis; the latter intensified even more with the loss of asmt1, which was the group most affected by the mutation. These malformations are signs of aging that have been observed in old zebrafish and associated with muscle degeneration [22]. Additionally, it has been shown that muscle structure and function are affected by chronodisruption, which is reflected in the physical activity of mice [12]. Because the alteration of melatonin production causes chronodisruption, we evaluated the influence of asmt1 and asmt2 mutations on the physical activity of adult zebrafish. Locomotor activity was more affected after the loss of asmt1. Interestingly, the group that showed the most malformations was likely due to the loss of muscle mass and strength, which we mentioned could be caused by melatonin deficiency.

The results of this study, along with the differential expression of asmt1 and asmt2 in zebrafish, suggest that they have different functions. Moreover, it is possible that the distinct effects observed between asmt1 and asmt2 mutations are not only due to tissue-specific expression but also stage-specific roles of melatonin. Given that *asmt2* is expressed mainly in peripheral tissues [9], its loss strongly impacts total melatonin levels and, consequently, the circadian behavior of larvae, as we observed. The greater impact on the larvae could suggest a role for ASMT2 during early development by influencing processes like cell proliferation, differentiation, and the establishment of circadian rhythms. In contrast, asmt1 is expressed in the pineal gland and the retina [9], and it is released into circulation in a circadian pattern, acting as a systemic chronobiotic signal. The loss of asmt1 may lead to chronic chronodisruption, which in turn affects motor activity and leads to the appearance of malformations in adult zebrafish. This could be due to the disruption of muscle homeostasis and the onset of aging signs, which are characteristic of chronodisruption [6]. Thus, asmt1 mutation may cause more subtle but long-term systemic effects, which are particularly relevant in adult physiology.

Thus, the development of a double knockout mutant for asmt1 and asmt2 would be necessary to determine how the total deficiency of melatonin influences zebrafish physiology. This is also applicable to the other melatonin synthesis enzyme, AANAT, of which zebrafish also have two genes (aanat1 and aanat2), with differences in expression and function [23,24].

## 4. Materials and Methods

### 4.1. Fish Maintenance

Adult zebrafish (Danio rerio) of the AB strain were obtained from ZFBiolabs S.L. (Madrid, Spain), while the mutant line sa11754 was supplied by the Sanger Institute (Cambridge, UK). Both lines were housed in a recirculating aquaculture system (Aquaneering Incorporated, Barcelona, Spain) at the University of Granada facilities. Fish were kept at a constant temperature of 28.5 ± 1 °C with a 14:10 h light/dark cycle (lights on at 08:00 h). All procedures related to animal care—including housing, feeding, reproduction, anesthesia, and euthanasia—were conducted following standardized protocols [25]. Adult zebrafish served as breeding stock, and larvae were raised in E3 medium under identical lighting conditions. All experimental procedures adhered to the guidelines established by the National Institutes of Health for the Care and Use of Laboratory Animals, the European Convention for the Protection of Vertebrate Animals for Experimental and Other Scientific Purposes (CETS No. 123), and Spanish legislation on animal experimentation (R.D. 53/2013). The study protocol received approval from the Andalusian Ethical Committee (Ref. #29/05/2020/068).

### 4.2. Generation and Genotyping of Zebrafish asmt Mutant Lines

The sa11754 line acquired from the Sanger Institute has a splice site mutation in exon 2 of the asmt2 gene located on chromosome 17, which affects amino acid 27 out of 348 and consists of a point mutation where G/A is changed. The zebrafish mutant line for asmt1 was generated through CRISPR/Cas9-mediated genome editing according to Moreno-Mateos et al. [26], producing a splice site mutation (G/A) in amino acid 55 in exon 2 of chromosome 9. A single guide RNA (sgRNA) was designed using the CRISPRscan algorithm to specifically target an optimal CRISPR sequence on exon 2 of the asmt1 gene. The chosen sgRNA (Appendix A) was transcribed in vitro using the AmpliScribe-T7 Flash Transcription Kit (Izasa Scientific, Madrid, Spain), and the Cas9 mRNA was generated from the pCS2-nCas9n-nanos 3′UTR plasmid (62542, addgene, Bethesda, MD, USA). One-cell-stage larvae were injected with 100 ng/μL of Cas9 and 30 ng/μL of sgRNA; phenol red was used as the injection marker.

Both mutant lines were crossed with AB WT zebrafish to obtain heterozygotes, and these were crossed with each other to obtain homozygotes. Therefore, the groups obtained were WT, asmt1^+/−^, asmt1^−/−^, asmt2^+/−^, and asmt2^−/−^. The primers for genotyping each mutant are listed in Appendix A.

### 4.3. Determination of Melatonin Concentration

Zebrafish larvae were collected at 5 days post-fertilization (dpf), all simultaneously at the end of the dark cycle, for the determination of melatonin levels using ultra-high-performance liquid chromatography coupled to tandem mass spectrometry (UHPLC–MS/MS) [27]. Melatonin extraction was performed with chloroform, followed by solvent evaporation and reconstitution in the mobile phase. A 25 µL aliquot of each sample was injected into the UltiMate 3000 UHPLC system (ThermoFisher Scientific, Madrid, Spain) equipped with a Hypersil GOLD C18 reverse-phase column (100 mm × 2.1 mm, 1.9 µm; ThermoFisher Scientific, Madrid, Spain).

Chromatographic separation was achieved using a mobile phase composed of water with 0.1% formic acid (solvent A) and acetonitrile with 0.1% formic acid (solvent B). A linear gradient elution was applied, increasing from 5% to 98% B over 7 min, maintained at 98% B for 2.1 min, and then returned to 5% B during a 0.9 min equilibration phase. The total run time was 10 min. The mobile phase flow rate was set at 0.4 mL/min, with autosampler and column temperatures maintained at 10 °C and 45 °C, respectively.

Melatonin detection was carried out using a Q-Exactive Focus Orbitrap mass spectrometer (ThermoFisher Scientific, Madrid, Spain) operating in positive electrospray ionization (ESI) mode and using selective ion monitoring (SIM). Data acquisition was managed with Xcalibur software v.4.1.31.9 (ThermoFisher Scientific, Madrid, Spain). The protonated melatonin ion (*m*/*z* 233.12845) was detected with a retention time of 3.98 min.

### 4.4. Sleep/Wake Rhythm Analysis

Zebrafish larvae at 5 days post-fertilization (dpf) were maintained in 48-well plates, with each well containing 1 mL of E3 medium. Locomotor activity was assessed as the total distance moved (in mm) per individual recorded using the ZebraBox tracking system (Viewpoint, Lyon, France) in live mode. The assay lasted 24 h under a light/dark cycle of 14 h light and 10 h dark, with larvae being acclimated to the testing conditions the day prior to the experiment. The activity detection threshold was set at 25, and inactivity was defined as movement below 0 mm/s. Measurements were taken every 30 min, and data were acquired using Zebralab software version 3.22.3.9. This assay was adapted from Doldur-Balli et al. [28].

### 4.5. Morphometric Analysis and Malformation Quantification

Phenotyping was conducted on 5 dpf zebrafish in order to evaluate larval development. Total length (μm), width (μm), eye length (μm), ear length (μm), and swim bladder length (μm) were macroscopically quantified.

The quantification of malformation was analyzed in 5 dpf larvae and 2-year-old adult zebrafish. Edema and tail and yolk abnormalities, the most common malformations in larvae, and scoliosis and malformation in the caudal tail, principle signs of aging in adult zebrafish, were macroscopically quantified.

### 4.6. Assessment of Locomotor Activity

Three-month-old adult zebrafish were individually placed in tanks filled with system water and acclimated over a period of four days. On the fifth day, behavioral assessments were performed. Locomotor activity was tracked for 20 min, following a 3 min habituation phase, using a digital video tracking setup that included a CCD camera connected to a computer [29]. The video recordings were analyzed using SMART 3.0 software (Panlab, Harvard Apparatus, Barcelona, Spain). For each fish, the total distance moved (cm), average speed (cm/s), peak speed (cm/s), and percentage of resting time were quantified.

### 4.7. Statistical Analysis

Statistical analyses were carried out using GraphPad Prism v. 8.0.1 software (GraphPad, Software, Inc.; La Jolla, CA, USA). Data are expressed as the mean ± S.E.M. An unpaired *t* test and one-way or two-way ANOVA with Tukey’s post hoc test were used to compare the differences between the experimental groups. A *p*-value of 0.05 was considered to be statistically significant.

## Figures and Tables

**Figure 1 ijms-26-03912-f001:**
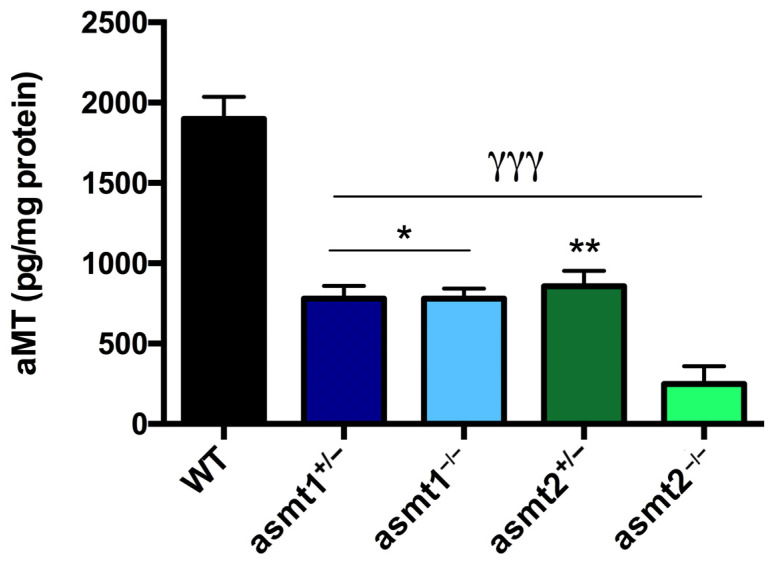
Melatonin concentration in zebrafish larvae. asmt1^+/−^, asmt1^−/−^, and asmt2^+/−^ showed a significant reduction of 50% compared to WT. asmt2 ^−/−^ revealed an even greater decrease compared to the WT and the other mutant lines. Data are presented as mean ± SEM (*n* = 5 larvae/group). γγγ *p* < 0.001 vs. WT; * *p* < 0.05 vs. asmt2^−/−^; ** *p* < 0.01 vs. asmt2^−/−^. One-way ANOVA with Tukey’s post hoc test.

**Figure 2 ijms-26-03912-f002:**
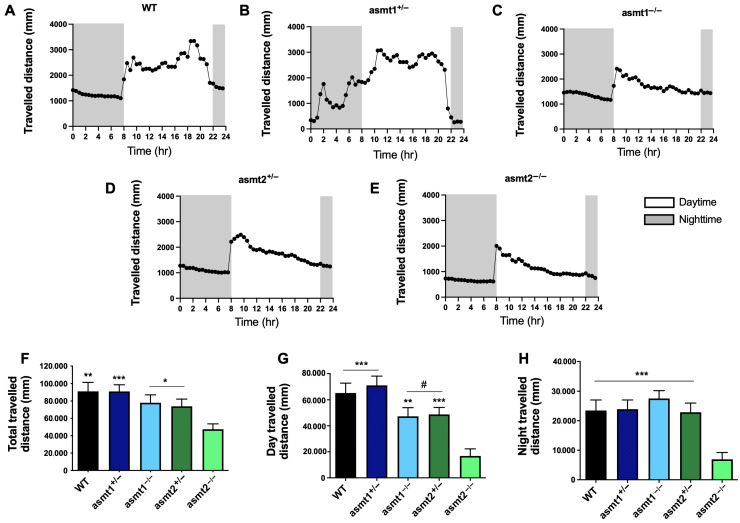
Total distance traveled during 24 h, 14 h light and 10 h darkness, by zebrafish larvae. (**A**) The WT group exhibited a normal activity pattern, with more activity during the day than during the night. (**B**) asmt1^+/−^ experienced an advance in the activity phase, even at night. (**C**) asmt1^−/−^, (**D**) asmt2^+/−^, and (**E**) asmt2^−/−^ did not show an activity rhythm. (**F**) The total distance traveled decreased significantly in asmt2^−/−^ compared to the other groups. (**G**) asmt1^−/−^ and asmt2^+/−^ significantly decreased the distance traveled during the day compared to asmt1^+/−^ and asmt2^−/−^ compared to the rest of the groups. (**H**) Nighttime activity was significantly lower in asmt2^−/−^ than in the WT and the other mutant lines. Data are presented as mean ± SEM (*n* = 12 larvae/group). # *p* < 0.01 vs. asmt1^+/−^; * *p* < 0.05 vs. asmt2^−/−^; ** *p* < 0.01 vs. asmt2^−/−^; *** *p* < 0.001 vs. asmt2^−/−^. Unpaired *t* test.

**Figure 3 ijms-26-03912-f003:**
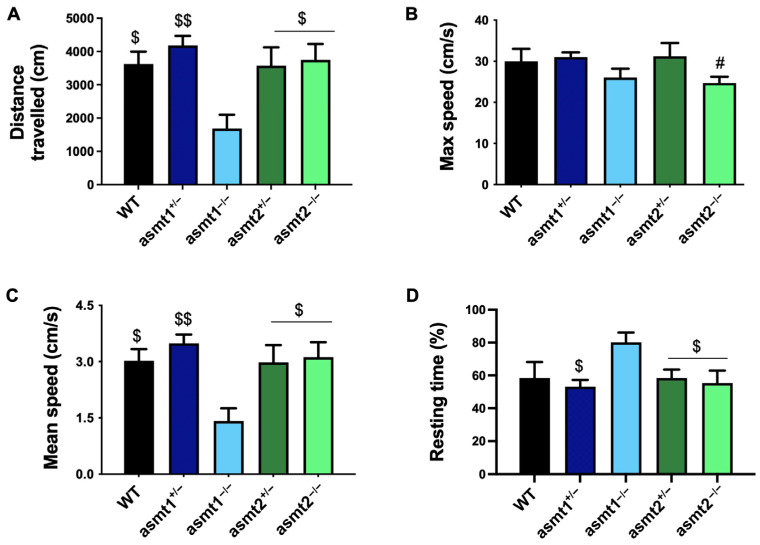
Locomotor activity recorded in adult zebrafish. (**A**) asmt1^−/−^ showed a significant reduction in the distance traveled compared to WT, asmt1^+/−^, asmt2^+/−^, and asmt2^−/−^. (**B**) Both asmt1^−/−^ and asmt2^−/−^ showed a slight decrease in maximum speed compared to WT and heterozygotes, but only asmt2^−/−^ was significant. (**C**) The mean speed only decreased significantly in asmt1^−/−^ compared to the other groups. (**D**) asmt1^−/−^ increased the resting time more than WT, asmt1^+/−^, asmt2^+/−^, and asmt2^−/−^. Data are presented as mean ± SEM (*n* = 5 larvae/group). # *p* < 0.05 vs. asmt1^+/−^; $ *p* < 0.05 vs. asmt1^−/−^; $$ *p* < 0.01 vs. asmt1^−/−^. Unpaired *t* test.

**Figure 4 ijms-26-03912-f004:**
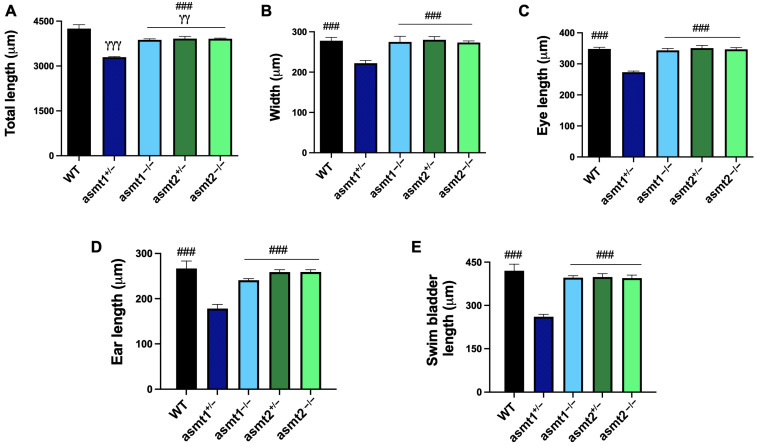
Morphometric aspects of zebrafish larvae. (**A**) The total length was significantly lower in asmt1^+/−^, asmt1^+/−^, asmt2^+/−^, and asmt2^−/−^ than in WT. The asmt1^+/−^ mutant line showed a significant reduction in (**B**) width, (**C**) eye length, (**D**) ear length, and (**E**) swim bladder length compared to WT and the other mutant groups. Data are presented as mean ± SEM (*n* = 5 larvae/group). γγ *p* < 0.01 vs. WT; γγγ *p* < 0.001 vs. WT; ### *p* < 0.001 vs. asmt1^+/−^. One-way ANOVA with Tukey’s post hoc test.

**Figure 5 ijms-26-03912-f005:**
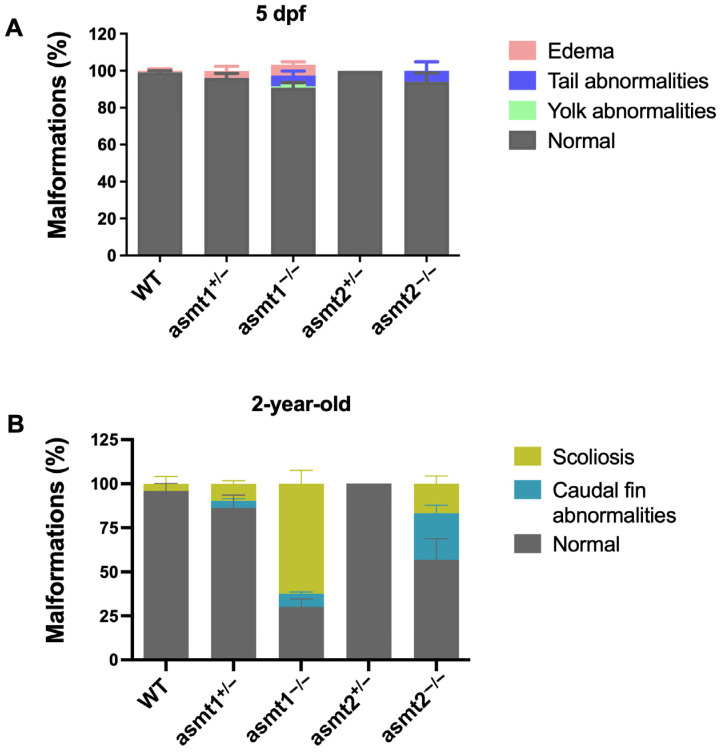
Malformations in 5 dpf zebrafish larvae and 2-year-old adult zebrafish. (**A**) Some asmt1^+/−^ zebrafish showed edema, while in asmt1^−/−^, all three types of malformation were found, and tail malformation was observed in asmt2^−/−^, contrasting with WT and asmt2^+/−^, which did not show any malformations. (**B**) Malformations were found to increase in the asmt1^+/−^ group, which were even greater in the homozygous lines. Scoliosis predominated in asmt1^−/−^ and caudal fin malformation in asmt2^−/−^. However, asmt2^+/−^ did not exhibit any malformations. Data are presented as mean ± SEM (*n* = 5 larvae/group).

## Data Availability

The datasets generated during and/or analyzed during the current study are available from the corresponding author (dacuna@ugr.es) upon reasonable request. Materials described in the manuscript will be freely available to any researcher to use them for noncommercial purposes.

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
