# Peer review of "Lack of asmt1 or asmt2 Yields Different Phenotypes and Malformations in Larvae to Adult Zebrafish"

_ijms, 2025, doi:10.3390/ijms26083912_

Round 1
Reviewer 1 Report
Comments and Suggestions for Authors
International Journal of Molecular Sciences (Manuscript ID: ijms-3570624), Comments to the Authors:
Title: Lack of asmt1 or asmt2 yields different phenotypes and malformations from embryos to adults zebrafish
Comments
The submitted manuscript focused on developing a mutant model for each asmt gene and analyzed their role in phenotypic aspects in zebrafish. The results showed that the loss of asmt2 affected melatonin concentration and consequently the sleep/wake rhythm in embryos, and the loss of asmt1 had a greater influence on the physical condition of adults. These data revealed well differential functions of melatonin depending on their site of production that may affect the developing of zebrafish.
I think the submitted paper can be accepted after the authors respond to the following comments:
- The abstract can be improved by providing numerical data. A statement like “melatonin concentration decreased much more significantly in asmt2 homozygous mutants” – can be improved by including how much was the decrease?
- The phrase “discover well differential functions” is unclear and should be rephrased.
- The rationale for studying asmt1 and asmt2 mutants in zebrafish was not clearly defined by the authors. The authors should provide a clear rational for their study.
- The authors should comment on why the asmt1 loss disproportionately affected adult physical condition despite less impact on melatonin levels?
- The authors described the presence of phenotypic differences but offered limited mechanistic explanations. Why the asmt2 loss caused a greater melatonin drop, and why the asmt1 loss disproportionately affected adult physical activity? Biochemical or molecular assays (e.g., gene expression, oxidative stress markers) could improve the significance of the paper.
Author Response
- The abstract can be improved by providing numerical data. A statement like “melatonin concentration decreased much more significantly in asmt2 homozygous mutants” – can be improved by including how much was the decrease?
Included.
- The phrase “discover well differential functions” is unclear and should be rephrased.
Rephrased.
- The rationale for studying asmt1 and asmt2 mutants in zebrafish was not clearly defined by the authors. The authors should provide a clear rational for their study.
Provided al the end of the introduction.
- The authors should comment on why the asmt1 loss disproportionately affected adult physical condition despite less impact on melatonin levels?
Commented in the discussion.
- The authors described the presence of phenotypic differences but offered limited mechanistic explanations. Why the asmt2 loss caused a greater melatonin drop, and why the asmt1 loss disproportionately affected adult physical activity? Biochemical or molecular assays (e.g., gene expression, oxidative stress markers) could improve the significance of the paper.
Commented in the discussion.
Reviewer 2 Report
Comments and Suggestions for Authors
In this study, the authors developed a mutant model for each asmt gene, and analyzed their role in phenotypic aspects in embryos and adult’s zebrafish. The authors suggested that the loss of asmt2 mainly affected melatonin concentration and consequently the sleep/wake rhythm in embryos, while the loss of asmt1 had a greater influence on the physical condition of adults.
Comments:
The reviewer has some concerns as follows:
- This is an interesting study. The manuscript is well-written. However, there are some concerns for data presentation:
(1) The so-called “embryo” in this study refers to the 5 dpf zebrafish used. This stage is consistent with the “larva” stage, so calling it “larva” may be more suitable for the actual growth situation.
(2) How about the survival rate for these mutation zebrafish models?
(3) Has the enzyme activity for melatonin synthesis (such as AANAT and ASMT) really changed?
(4) How about the changes in melatonin concentration in adult fish with WT and mutation?
(5) In Figures 1 and 2-F-H, is there a statistically significant difference between the asmt2+/- and asmt2-/- groups?
(6) In Figure 5A and B, the statistical analysis seems to be lacking.
- The references cited in this manuscript are appropriate relevant to this research. However, this manuscript contains several self-citations (9 references), please reduce them carefully.
- Overall, this manuscript needs a revision before it can be accepted.
Author Response
- This is an interesting study. The manuscript is well-written. However, there are some concerns for data presentation:
(1) The so-called “embryo” in this study refers to the 5 dpf zebrafish used. This stage is consistent with the “larva” stage, so calling it “larva” may be more suitable for the actual growth situation.
Changed.
(2) How about the survival rate for these mutation zebrafish models?
We did not observe any changes in survival rate.
(3) Has the enzyme activity for melatonin synthesis (such as AANAT and ASMT) really changed?
We have confirmed the absence of the ASMT 1 or 2 gene by sequencing, and we have observed a decrease in melatonin production, but we have not measured the enzyme activities.
(4) How about the changes in melatonin concentration in adult fish with WT and mutation?
The melatonin concentration has not been measured in the adult fish.
(5) In Figures 1 and 2-F-H, is there a statistically significant difference between the asmt2+/- and asmt2-/- groups?
Yes, both the concentration of melatonin and the activity are significantly reduced in asmt2-/- group compared to asmt2+/-.
(6) In Figure 5A and B, the statistical analysis seems to be lacking.
Statistical data of these figures is reported the Supplementary table S2 and S3.
- The references cited in this manuscript are appropriate relevant to this research. However, this manuscript contains several self-citations (9 references), please reduce them carefully.
This point has been addressed.
- Overall, this manuscript needs a revision before it can be accepted.
The manuscript has been revised and some parts rewrited.
Reviewer 3 Report
Comments and Suggestions for Authors
Introduction. Are there any other works with mutations in ASMT in other organisms?
Did the authors perform a double knockout mutant for asmt1 and asmt2? This article can present these results. MDPI journals do not restrict the manuscript length. This work would be beneficial if the authors included results with this double knockout mutant.
The authors conducted behavioral experiments. Are there any other behavioral assays to support the results regarding the sleep/wake rhythm?
In Section 4.3, did the authors validate the method for melatonin quantification?
Line 83: There are two points at the end of the sentence.
In sections 4.4 and 4.6, were these assays previously performed by other research groups? A reference for each method should be included.
Discussion. This section needs improvement. The results presented here are of relevance.
Author Response
Introduction. Are there any other works with mutations in ASMT in other organisms?
There are not many studies on asmt mutations in other animal models. There are some studies, but they are not closely related to the topic of our article, which is why we had not mentioned them:
-Weina Liu et al., Commun Biol, 2023 (https://doi.org/10.1038/s42003-023-05520-8) demonstrate the role of asmt in the microbiota in females.
-Teng Ma et al., J Pineal Res, 2017 (https://doi.org/10.1111/jpi.12406) study the enrichment of melatonin in sheep milk.
-Wenbin Liu et al., Biosci Rep, 2022 (doi: 10.1042/BSR20220800) observe that the absence of asmt leads to depression.
Did the authors perform a double knockout mutant for asmt1 and asmt2? This article can present these results. MDPI journals do not restrict the manuscript length. This work would be beneficial if the authors included results with this double knockout mutant.
The development of the double knockout is planned for the upcoming experiments, as it is important for understanding the role of ASMT1 and ASMT2 in zebrafish.
The authors conducted behavioral experiments. Are there any other behavioral assays to support the results regarding the sleep/wake rhythm?
We only conducted the sleep/wake study in order to determine whether there was an alteration in the circadian system. In the literature, there are other behavioral assays used to study sleep in zebrafish, such as stimulus response or sleep deprivation (Chiu, C et al., Front. Neural Circuits 2013, https://doi.org/10.3389/fncir.2013.00058).
In Section 4.3, did the authors validate the method for melatonin quantification?
Yes, we designed the method in our laboratory, and it was validated in the paper by Lozano-Lorca et al., 2022. It has since been implemented in our group as the standard method for measuring melatonin, adapted to each type of sample (in this case, zebrafish larvae). I included the reference in the manuscript.
Line 83: There are two points at the end of the sentence.
Corrected.
In sections 4.4 and 4.6, were these assays previously performed by other research groups? A reference for each method should be included.
Included.
Discussion. This section needs improvement. The results presented here are of relevance.
The Discussion has been revised accordingly.
Round 2
Reviewer 1 Report
Comments and Suggestions for Authors
International Journal of Molecular Sciences (Manuscript ID: ijms-3570624), Comments to the Authors:
Title: Lack of asmt1 or asmt2 yields different phenotypes and malformations from embryos to adults zebrafish
Comments
After reading the revised version of the manuscript, I think it can be accepted.
Author Response
Comments: After reading the revised version of the manuscript, I think it can be accepted.
Thanks very much for your comments.
Reviewer 2 Report
Comments and Suggestions for Authors
This revised manuscript has a great improvement and the reviewer has no further comments.
Author Response
Comments: This revised manuscript has a great improvement and the reviewer has no further comments.
Thanks very much for your comments.
Reviewer 3 Report
Comments and Suggestions for Authors
The authors should state in the introduction section that there are not many studies on asmt mutations closely related to the topic of this work.
The role of asmt in other species should be mentioned in the discussion section.
Author Response
-The authors should state in the introduction section that there are not many studies on asmt mutations closely related to the topic of this work.
Information included in theIntroduction section, lines 80-81.
-The role of asmt in other species should be mentioned in the discussion section.
Included in the Discussion section, lines 223-230.